# Measuring Attitudes toward Suicide Prevention among Occupational Staff Frequently Exposed to Suicidal Individuals: Psychometric Evaluation and Validation

**DOI:** 10.3390/ijerph18084001

**Published:** 2021-04-10

**Authors:** Inga-Lill Ramberg, Sebastian Hökby, Linda Karlsson, Gergö Hadlaczky

**Affiliations:** 1National Centre for Suicide Research and Prevention, Department of Learning, Informatics, Management and Ethics, Karolinska Institute, 17177 Solna, Sweden; sebastian.hokby@ki.se (S.H.); linda.karlsson@ki.se (L.K.); gergo.hadlaczki@ki.se (G.H.); 2National Centre for Suicide Research and Prevention, Centre for Health Economics, Informatics and Health Services Research, Stockholm Health Care Services, Karolinska Institute, 17177 Solna, Sweden

**Keywords:** suicide prevention, attitudes, staff, SEM, CFA, validity, reliability, questionnaire

## Abstract

As the attitudes of healthcare staff are thought to influence the quality and effectiveness of interventions targeting patients’ suicide risk, attitudes are often used as an outcome in the evaluation of suicide-preventive training. Due to various problems related to the validity and reliability of commonly used scales, there is a lack of overall agreement on how to measure these attitudes. Confirmatory factor analysis (CFA) was used to cross-validate previously used models and to investigate new models to measure professionals’ attitudes toward work with suicidal individuals and to test the longitudinal stability of the models by analyzing new sets of data. The population in the first study consisted of a heterogenous group of 1350 professionals who managed suicidal individuals relatively frequently. The second study included 640 professionals. The results of the cross-validation of previous models were described and a new questionnaire measuring attitudes toward suicide prevention, suicidal individuals, and organizational-facilitated self-efficacy (OSAQ-12) was presented. The three presented models retained a good fit and were stable over time. Valid and reliable measurement models that measure aspects of attitudes toward suicide are a prerequisite for conducting both cross-sectional and intervention studies.

## 1. Introduction

Measuring attitudes toward suicide and suicide prevention has been an endeavor of many researchers in the field of suicidology since the 1980s. Interventions aiming to improve attitudes toward suicide prevention are common elements of multi-component suicide preventive strategies [1,2] as the stigma surrounding suicide in the population could be a perceived hindrance for preventing suicidal acts. Attitudes toward suicide also play an important role in the willingness to implement suicide preventive interventions [3], as well as in facilitating the acceptance and effect of these interventions. Furthermore, the stigma surrounding suicide might influence the quality and effectiveness of care and, consequently, the patient’s suicidality. Due to the latter reason, staffs’ attitudes toward suicide is often used as an outcome measure when training in suicide prevention is evaluated. There is also a practical reason for the use of scales measuring attitudes. Training in suicide prevention is usually performed in smaller groups or in settings that make it difficult to measure direct effects on how the course knowledge influences the professional’s behavior toward suicidal individuals [4]. To alleviate this problem, intermediary outcome measures such as attitudes are often used instead. The rationale is that the distress of a suicidal individual can be counterbalanced by positive attitudes from health care staff, thus having a preventive effect on suicidal behaviors [5,6,7,8,9,10,11]. Although attitudes are generally considered as being relatively stable [12,13], it appears that the impact of being sufficiently trained to work with suicidal individuals is a more important predictor than age, gender, and work experience on attitudes concerning work with these individuals [14]. However, despite the attitudes toward suicide and its prevention being frequently used as outcomes, there is a lack of agreement on how to measure these attitudes. There are a variety of questionnaires available, but few have been investigated appropriately with regard to their psychometric properties.

Three often-used instruments that aim to investigate a broad aspect of attitudes in the general population or in subpopulations include the Suicide Opinion Scale (SOQ) by Domino et al. [15], the suicide attitude questionnaire (SUIATT) by Diekstra and Kerhof [16], and Attitudes towards suicide (ATTS) by Renberg and Jakobsson [17]. Although these scales have been used in several studies (see below), there is no consensus on the number of factors or the factor models, which is especially important when the study results are compared. The lack of consensus could in fact be a symptom of problems regarding validity and reliability.

In a systematic review of studies on the healthcare professionals’ knowledge of and attitudes toward caring for suicidal people, Boukouvalas et al. [18] found 27 cross-sectional studies and 19 intervention studies. In 10 of the cross-sectional studies, the authors constructed a questionnaire for their specific study, while four used ATTS, and another four used the Understanding of Suicide Attempt Scale (USP), which was developed by Samuelsson et al. [19]. Others used scales that are less common, such as the Cuestionario de Creencias Actitudinales sobre el Comportamiento Suicida (CCCS-18) by Ruiz-Hernández et al. [20]; the Stigma of Suicide Scale (SOSS) by Batterham et al. [21]; and the Scale of Public Attitudes about Suicide (SPAS) by Li XY et al. [22].

Seven of the 19 intervention studies found by Boukouvalas et al. [18] were performed with ad hoc questionnaires constructed by the authors for the purpose of the study. Five of the studies used the Suicide Behavior Attitude Questionnaire (SBAQ), which was developed by Botega et al. [23], two used the USP-scale, and one used the SOQ. Examples of less frequently used scales are the Attitudes to Suicide Prevention Scale [ASPS] by Herron et al. [24] and the Depression Stigma Scale (DSS) by Griffiths et al. [25].

Some of the measurement properties of the scales used in these cross-sectional and intervention studies were analyzed using exploratory factor analysis (EFA) or reliability tests such as Cronbach’s alpha. Cronbach’s alpha is widely used to scrutinize the internal consistency and homogeneity of scales. However, not without criticism, for instance, Cho and Kim [26] argued that it does not give reliable information on the internal consistency and reliability of a measurement instrument. Instead, they recommend that structural equation modeling (SEM) should be used to analyze the dimensionality of a scale. EFA, as well as Cronbach’s alpha, is also limited by its data-driven approach. Thus, researchers do not make any specifications beforehand regarding the number of latent factors or to the pattern of relationships between the common factors and the items. EFA also has the disadvantage of often producing results that are difficult to interpret and that the interpretations could differ between researchers [27,28]. EFA is therefore best used early in the process of scale development and construct validation but should not be interpreted as a method of verification [28].

To achieve theory-based and more robust instruments, confirmatory factor analysis (CFA) is arguably preferred. CFA starts with the conceptual specification of a measurement model for each concept and forces one to be explicit about the measurement hypothesis. This technique was used to cross-validate the USP-scale [29]. USP originally consists of 11 items that are subjected to an item analysis [30]. Item analysis is a technique that evaluates the effectiveness of items in a scale and could, like EFA, be useful for a descriptive and exploratory procedure but should not be regarded as a reliable and valid verification of a scale. Nine of the 11 items in the USP-scale were included in another Swedish study on psychiatric staff’s attitudes toward caring for suicidal patients. When these items were tested with CFA, Aish et al. [29] found that the data did not fit the one-factor model suggested by Samuelsson et al. [19].

Recently, the psychometric properties of some other attitude scales have been scrutinized using CFA. Cwik et al. [31] analyzed a German translation of the Cognitions Concerning Suicide Scale [CCSS] by Biblarz et al. [32]. They found that 18 of the 20 original items measured on a six-point Likert scale retained a three-factor solution (“right to commit suicide,” “interpersonal gesture,” “resilience”), which they considered having a good fit, measured by some of the fit parameters, and only a reasonable fit measured by others. However, they concluded that the generalizability could be questioned as the participants were predominately young female students.

VanSickle et al. [33] designed a questionnaire with 35 items with a five-point Likert scale to measure attitudes in a military population. An EFA produced a four-factor model [Individual based rejection versus acceptance, Psychache versus pathological, Unit-based acceptance versus rejection, Moral versus Immoral] with 34 items. When testing each factor in this model with CFA, two more items were excluded. The authors considered the four measurement models to have a good fit. It is noteworthy that each factor was tested separately with CFA. Consequently, the four-factor model produced by EFA was not confirmed. Instead, the results suggested that the outcome value(s) of each factor should be analyzed and interpreted separately from the other three.

Ji et al. [34] evaluated a Korean version of the Attitudes towards suicide (ATTS) scale by Renberg and Jacobsson [17]. They found that instead of the original scale that consisted of a 10-factor model with 34 items, their EFA produced an 11-factor model (Acceptability, Acceptability related incurable disease, Preventability, Tabooing, Unpredictability, Normal-common, Aging, Incomprehensibility, Suicidal process, Non-communication and Relation-caused) with 32 items. This model was found acceptable in terms of both internal consistency and test–retest reliability. However, a test with CFA showed that the measurement model did not fit the data, which means that the Attitudes towards suicide (ATTS) scale cannot be regarded as a valid and reliable scale. The fact that most of the factors consisted of only two or three items could be considered as a warning signal.

Sandford et al. [35] evaluated the Attitude to Suicide Prevention Scale [ASPS] by Herron et al. [24] with EFA and CFA. Tests with EFA revealed that the 14 items formed a two-factor model when two of the items were eliminated. A second EFA revealed that a one-factor model had a better fit when the two EFA models were compared using parallel analysis with ANOVA. The two models were then explored with CFA, but neither of the models were good fits. Thus, the items of ASPS should not be treated as being part of a coherent scale, only as single items.

The results of these studies show the advantage of CFA over EFA. However, there is a lack of information in all the above studies concerning the fit of the models, which make it impossible for the reader to fully evaluate the results of the studies. In a model evaluation with CFA, it is strongly advised by, for instance, Brown [28], Steiger and Lind [36], Jöreskog and Sörbom [37], Bentler [38], Hu and Bentler [39], and Hayduk et al. [40] that the overall fit of a model is described by the absolute fit by the chi-square value along with its degrees of freedom and *p*-value and the standardized root-mean-square residual (SRMR), the parsimony fit by the mean square error of approximation (RMSEA), and the comparative fit by, for example, the comparative fit index (CFI). The local fit of the model should be described by the parameter estimates such as factor loadings and error variances. It is also recommended that the path diagram is presented [28,41,42]. The reason why reporting a variety of fit indices is necessary is that different indices reflect different aspects of the model fit [43]. Furthermore, in none of these studies, the authors reported how they analytically managed the fact that the items were ordinal. With categorical data, maximum likelihood (ML) should not be used in the estimations, as this could jeopardize the results, and in other cases, produce incorrect test statistics and standard errors [28,44,45].

In the present study, we focused on models to measure professionals’ attitudes toward work with suicidal individuals with the aim of establishing the psychometric properties of an instrument created for the purpose of evaluating the effects of suicide-preventive training on attitudes.

The aim of this study was to [1] cross-validate the psychometric properties of measurement models used in psychiatric care for attitudes toward work with suicidal persons (Job clarity and Job confidence) and attitudes toward the possibility to prevent suicides [14] on a heterogenous population; [2] investigate whether other items concerning attitudes toward suicidal persons and towards work with these persons could form new measurement models; and [3] test the reliability of these measurement models on a new dataset.

## 2. Methods

### 2.1. Participants

The sample consisted of a variety of professionals (clinical and nonclinical staff) that had registered for a one-day course in suicide prevention, held by the Swedish National Centre for Suicide Research and Prevention (NASP) during the period of 2014–2017. Two weeks before the course took place, an electronic questionnaire containing 58 items was sent to the N = 3346 registered course participants. The number of persons that completed the questionnaire was n = 2422 [72.4%]. One and a half years after the course took place, the same electronic questionnaire was sent to the n = 1773 course participants that were still reachable. The number of completed questionnaires was n = 1082 [61%].

The invitation to the course was sent to a wide range of institutions in the Stockholm region whose employees might potentially encounter or communicate with suicidal individuals. Thus, the population was very heterogenous, representing, for example, social welfare, primary and hospital care (both somatic and psychiatric), schools, police, churches, fireworks, employment, and insurance services. Although one requirement for participation in the course was the experience of having worked with suicidal persons, the amount of experience participants had varied widely. Participants that did not report working with suicidal individuals at least once a month during the past six months were excluded from the analyses. Of the 1350 included respondents in the first study, 84% were females, 15.5% were males, and 0.5% reported “other”. The mean age was 43.95 (SD 11.555). In the second study, 648 (83.5% females, 16.4% males, 0.2% “others”) respondents were included. The mean age was 47.38 (SD 11.144).

### 2.2. Items

This study analyzed 22 items that aimed to examine various aspects of attitudes toward work with suicidal individuals (see Table 1 and Table 2). The responses to all items were scored using a Likert scale with numbers of 1 (do not agree at all), 2 (hardly agree), 3 (almost agree), and 4 (agree completely). The values for the negative items were reversed, i.e., the higher value, the more positive the attitudes.

## 3. Procedures

### 3.1. The First Study

#### 3.1.1. Cross-Validations

Confirmatory factor analysis (CFA) was performed to cross-validate three measurement models that were previously found to meet the criteria for good fitting models in psychiatric populations (see [14]), Job clarity, Job confidence, and Attitudes towards prevention. The cross-validations were performed on the total sample of 1350 respondents that had reported working with suicidal individuals during the past six months.

#### 3.1.2. Tests of the Hypothesized Measurement Models

To test the hypothesized measurement models of Job support and Attitudes toward suicidal persons, the N = 1350 respondents that had reported working with suicidal individuals during the past six months were randomly allocated into two test groups (Group G1: n = 698; Group G2: n = 652). G1 was used to explore the data with Exploratory factor analysis (EFA), and G2 to test if the hypothesized models or the results of the EFA could be confirmed in a confirmatory factor analysis (CFA) and, if not confirmed, whether there were any alternative models providing a good fit. The total sample of N = 1350 was finally used to confirm or reject the results of the CFA in G2. Here, it is worth noting that large sample sizes are generally more likely to produce significant *p*-values. Randomly splitting the dataset into parts (G1 and G2) directly downregulated the risk of type-2 errors (*p* < α) related to this sample size bias.

### 3.2. The Second Study

#### Reliability Tests

The measurement models that were found to have a good fit in the first study were tested using CFA on the data from the 648 occupational staff frequently exposed to suicidal individuals that also responded to the second questionnaire.

## 4. Data Analyses

IBM SPSS Statistics 25 was used to randomly split the total sample into two test groups.

LISREL 9.30 [47] was used to perform the exploratory factor analysis (EFA) for ordinal data. The cutoff for eigenvalues was set to 1.0.

Confirmatory factor analysis (CFA), which is a type of structural equation modeling (SEM), was performed using LISREL 9.30 [47]. This technique enables one to test whether relationships expected on theoretical grounds appear in the data [48]. This means that the theoretical models must be defined beforehand (see Table 1 and Table 2). As the data consisted of ordinal data without metric properties, i.e., they are not continuous and thus not measured on a continuum or a scale, PRELIS [47] was used to estimate the polychoric correlations and their asymptotic covariance matrix when testing the above one-factor models [28,44].

In scrutinizing the results of the CFA, the local fit, as well as the overall fit, must be considered. The overall fit is assessed by the chi-square test, root-mean-square error of approximation (RMSEA), comparative fit index (CFI), and standardized root mean square residual (SRMR). The local fit concerns the validity and reliability of the individual items, where the validity estimates the loading of the observed variable (items) on the latent variable (i.e., Job clarity). The reliability (R^2^) measures the proportion of the unique variance in an item. The items can also be evaluated by examining their statistical significance. Z-values of ±1.96 or greater are statistically significant [28].

By default, the path diagrams generated by LISREL are based on the Satorra–Bentler scaled chi-square test [49]. This chi-square test is recommended when data are ordinal as it takes nonnormality into account [44].

A “good” model fit is conventionally defined by *p*-values larger than alpha (>0.05). There is a wide range of studies that have tried to find the cut-off criteria for RMSEA, CFI, and SRMR [36,39,50,51,52] (see Table 3 for an overview).

## 5. Results

### 5.1. Cross-Validations

#### 5.1.1. Construct A: Job Clarity

As can be seen from Figure 1, the confirmatory factor analysis on the total group shows that the χ^2^ value [8.28] is too high relative to its degrees of freedom (DF = 2). However, the RMSEA value indicates a reasonable fit, and the values for CFI and SRMR indicate a good fit. The high chi-square value is probably due to the poor local fit of the item for “Different superiors have varying views on how and what I shall do in work with suicidal individuals” (DIFFVIEW), as its loading (0.28) and unique variance (0.08) (not shown in Figure 1) take on low values, while still reaching statistical significance.

I know what is expected of me in work with suicidal individuals (KNOW); Different superiors have varying views on how and what I shall do in work with suicidal individuals (DIFFVIEW); I get clear and good instructions concerning management of care of suicidal individuals (CLEAR); I the lack knowledge and information about what is important in work with suicidal individuals (LACKINFO).

The fit indices were χ^2^_[df = 2]_ = 8.275, *p* = 0.0160, RMSEA 0.0668, *p* = 0.166, [CI 0.0362–0.102], CFI = 0.994, and SRMR = 0.0223

#### 5.1.2. Construct B: Job Confidence

The model for Job confidence did not fit the data. The chi-square *p*-value was <0.05 and the value of RMSEA was 0.147. The item “I have no one with whom I can share the responsibility for the suicidal individuals” (ALLRESP) seemed to contribute to the poor model fit as the correlation matrix shows that this item is poorly correlated with “The division of responsibilities for risk assessment is clear and distinct” (CLEARRES) and “I feel confident when working with suicidal individuals” (CONF). As the model consists of only four items, which is the minimum for a one-factor model to be over-identified, the model was rejected.

#### 5.1.3. Construct C: Attitudes toward the Possibility to Prevent Suicides

The results of the cross-validation of the measurement model for the construct Attitudes towards prevention are presented in Figure 2. The model has a good fit, χ^2^_[df = 2]_ = 2.883, *p* = 0.2366, RMSEA = 0.0514, *p* = 0.405 (CI 0.0205–0.0880), CFI = 0.999, SRMR = 0.0139. As can be seen in the figure, the model is somewhat imbalanced as the loadings of the indicators differ from a high loading for NODIFF (0.89) and a lower loading for PREVSUI (0.56). Thus, PREVSUI has a somewhat lower validity and reliability R^2^ = 0.31 than the other indicators.

It is possible to prevent suicides (PREVSUI); It makes no difference what is done for suicidal individuals—they succeed sooner or later anyway (NODIFF); If people really want to kill themselves, they will succeed despite receiving the best treatment (SUCCEED); Once people have made up their minds to commit suicide, you cannot stop them (CANNOTST).

#### 5.1.4. Summary of Cross-Validation Results

In summary, the cross-validations show that the only model with a good fit was Attitudes towards prevention. The fit indices for the construct Job clarity varies from a good fit (CFI and SMRS) and a reasonable fit (RMSEA) to a bad fit (chi-square). Finally, the measurement model Job confidence did not fit the data.

### 5.2. Testing New Hypothesized Measurement Models

#### 5.2.1. Construct D: Job Support

Five items were originally designed to measure Job support. However, the model with all five items included did not fit the data in subgroup G2. The *p*-value of the Satorra-Bentler chi-square was 0.001 and the RMSEA was 0.21.

As the items of the constructs Job support and Job confidence all measure various aspects of work-related routines, the items were analyzed together using exploratory factor analysis (EFA) for ordinal data on subsample G1, see Table 4.

As can be seen from Table 4, the EFA suggests a one-factor model where SUFFSUPP has the highest loading (0.902) and NOTKNOW the lowest (0.367). However, this one-factor model did not have a good fit [χ^2^ =342.422, df = 2, *p* = 0.000]. Guided by the Modification Indices and the Correlation Matrix, indicators were removed in the following order: UNCLEAR, NOTKNOW, ALLRESP, ASKHELP, SUFFRES. The remaining four items formed a good fit for subgroup G2 (Table 5) and for the total sample (Figure 3). As can be seen from Table 5 and Figure 3, the validity is high for all items, although it varies from 0.65 for CONF to 0.90 for SUFFSUPP for subgroup G2 and from 0.69 to 0.92 for the total sample.

The fit indices for the total sample are χ^2^_[df = 2]_ = 1.653, *p* = 0.4377, RMSEA = 0.0277, *p* = 0.782, (CI 0.00–0.0678), CFI = 1.00, and SRMR = 0.00805.

#### 5.2.2. Construct E: Attitudes toward Suicidal Persons

The five items that aimed to measure Attitudes towards suicidal persons did not fit the data. However, when the item NOINTENT was removed, the four remaining items formed a model with a reasonable fit for subgroup G2 (Table 6) and a good fit for the total sample (Figure 4). As can also be seen from Table 6 and Figure 4, the loadings of the items vary from 0.52 to 0.85 in subgroup G2 and from 0.55 to 0.87 in the total sample. It was the item MISUSERS that had the lowest validity and reliability while the item TIME had the highest.

The fit indices for the total sample were χ^2^_[df = 2]_ = 3.496, *p* = 0.1741, RMSEA = 0.0549, *p* = 0.342 [CI 0.0239–0.0917], CFI = 0.999, and SRMR = 0.0145.

### 5.3. Reliability Tests of Job Clarity, Organization-Facilitated Self-Efficacy, Attitudes toward Prevention, and Attitudes towards Suicidal Persons

The reliability tests were performed on the population of the second wave of the study. The results are presented in Figure 5, Figure 6, Figure 7 and Figure 8.

The model did not fit the data, as the *p*-value of the Santorra–Bentler scaled chi-square was 0.0013 and RMSEA = 0.145, so it was rejected.

This model was found to have a good overall fit (χ^2^_[df = 2]_ = 1.753, *p* = 0.4162, RMSEA = 0.0422 (CI 0.0–0.102), *p* = 0.490, CFI = 1.00, SRMR = 0.0113), as well as a good local fit.

The model had a good overall fit (χ^2^_[df = 2]_ = 1.136, *p* = 0.5668, RMSEA = 0.0430 (CI 0.0–0.102), *p* = 0.481, CFI = 1.00, SRMR = 0.0126), as well as a good local fit.

This model had a good overall fit (χ^2^_[df = 2]_ = 0.931, *p* = 0.9545, RMSEA = 0 (CI = 0.0–0.0441), *p* = 0.963, CFI = 1.00, SRMR = 0.00344), as well as a good local fit.

## 6. Discussion

One purpose of this study was to cross-validate three measurement models concerning attitudes toward work with suicidal individuals: (a) Job clarity and (b) Job confidence and the possibility to prevent suicidal acts; (c) Attitudes towards prevention, identified in a previous study by Ramberg and colleagues [14] set in a psychiatric population. The current sample included a more heterogenous group of professionals who, through their occupational role, were exposed to suicidal individuals relatively frequently. The second purpose of the study was to examine whether additional items could form new measurement models related to (d) attitudes toward suicidal persons, and toward work with these persons and (e) job support. Finally, the third purpose was to test the reliability of these measurement models on data using a follow-up study, as stability over time is a prerequisite for the possibility to evaluate intervention effects.

### 6.1. Cross-Validations

The three measurement models that were cross-validated were the constructs of (a) Job clarity, (b) Job confidence, and (c) Attitudes toward prevention. Of these three models, only Attitudes toward prevention formed a good overall fit (chi-square *p* = 0.24, RMSEA = 0.05, CFI = 0.999, SMRS = 0.01). The model was also reliable when it was tested on the population in the follow-up study (chi-square *p* = 0.57, RMSEA = 0.04, CFI = 0.1.000, SMRS = 0.01). The replication of these findings indicates that the attitudes encompassed by subscale C is applicable to both psychiatric settings and more general occupational settings. The validity and the reliability of the indicators were also acceptable in both studies, although they varied between items.

The construct of Job clarity did not meet the criterion of a good fit, with respect to the *p*-value of the chi-square (*p* = 0.02) when it was tested on the population in the first study. However, the RMSEA has a reasonable fit (0.07) according to Browne and Cudeck [50] who recommend a cutoff of 0.08. Job clarity meets this standard, and the stricter recommendations of near to 0.06 proposed by Hu and Bentler [39]. Furthermore, Job clarity values of CFI (0.99) and SRMR (0.02) also indicated a good fit. However, when it was replicated in the follow-up study, it did not fit at all.

This construct seems to perform better when applied to psychiatric staff compared to heterogenous occupational groups [14]. This discrepancy could be caused by the item “Different superiors have varying views on how and what I shall do in work with suicidal individuals” which, despite statistical significance, had a low validity (0.28) and reliability (0.08) in the first study and 0.34 and 0.12, respectively, in the follow-up study. The samples used in this study were drawn from a population that encompasses a large variety of employers, which are probably less stringent and hierarchical with regard to the management of suicidal individuals. In occupational categories where the management of suicidality is more infrequent or otherwise more peripheral to the staff, one would anticipate less refined routines and, therefore, also different interpretations of the questionnaire items. Moreover, this measurement model emanates from a set of items that were adapted from the Karasek–Theorell demand-control model [53] and designed to be particularly applicable to health-care organizations [46]. To conclude, there is a need for further investigations to see if this model could be recommended for investigations of attitudes among psychiatric staff.

The cross-validation of the measurement model for the construct Job confidence did not fit the data and was therefore rejected. According to the correlation matrix, it seems as if the model was miss-specified, considering that the item “I have no one with whom I can share the responsibility for the suicidal individuals” is poorly correlated to the items “The division of responsibilities for risk assessment is clear and distinct” and “I feel confident when working with suicidal individuals.” It might be that the wording of the item “I have no one with whom I can share the responsibility for the suicidal individuals” is incompatible with a heterogenous population, compared to psychiatric staff [14]. As the model consists of only four items, it could not be re-specified.

### 6.2. Tests of New Measurement Models

Two new measurement models, not previously researched, were hypothesized and tested with CFA. The proposed model for Attitudes toward suicidal persons consisted of four items and the overall fit (chi-square *p* = 0.17, RMSEA = 0.05, CFI = 0.999, SRMR = 0.01), as well as the local fit of the model, was good, although the loadings/validity and the reliability varied between items.

However, we did not find a good fit for the last model, which included five items and intended to measure Job support. As the items of Job confidence and Job support all measure aspects of work-related routines, an exploratory factor analysis (EFA) for ordinal data was performed to test the factor structure of the nine items. The analysis resulted in a one-factor model. However, when this model was tested with CFA, it did not form a good fit. This shows the drawback of EFA over CFA and the disadvantage of purely relying on the results of EFA in this kind of research context, as EFA is an exploratory technique [28]. Based on the Modification Indices and the Correlation Matrix, the items UNCLEAR, NOTKNOW, ALLRESP, UNCLEAR, and SUFFRES were excluded one by one. The four remaining items (9. “The division of responsibilities for risk assessment is clear and distinct,” 10. “I feel confident when working with suicidal individuals,” 11. “The co-operation concerning the suicidal individuals is well functioning,” 12. “I have enough support in work with suicidal individuals”) formed a model with a good overall fit, which was called Organization-facilitated self-efficacy (chi-square *p* = 0.44, RMSEA = 0.03, CFI = 1.00, SRMR = 0.01). The local fit was also good, although the loadings/validity and reliability differed among items.

Both the models for Attitudes toward suicidal persons and the model for Organization-facilitated self-efficacy continued to retain a good overall and local fit when they were tested on the population in the follow-up study.

### 6.3. Summary and Scale Development

Three good-fitting measurement models, which were valid and reliable over time, were found when they were analyzed with CFA on a heterogenous sample of individuals, whose unifying characteristic was their exposure to suicidal individuals during the past six months. These models measure attitudes toward prevention, attitudes toward suicidal persons, and organizational-facilitated self-efficacy. The final questionnaire and its included items are presented below (Table 7).

The responses to all items were scored using a Likert scale with values of 1 (do not agree at all), 2 (hardly agree), 3 (almost agree), and 4 (agree completely).

### 6.4. Limitations and Future Research

According to Brown [28], one of many advantages of CFA is that one can test a parsimonious solution by indicating the number of factors and the pattern of factor loadings. In contrast to EFA, CFA allows for the specification of relationships among the item uniqueness (error variances). Nevertheless, although an SEM-model has been verified by the data, it does not mean that it has been proven unequivocally true. It just means that it has not been falsified, but there may be competing models [54]. As the focus in this study was to test the validity and reliability of existing and hypothesized measurement models, we did not have competing nonfalsified models to compare with. This could indeed be an objective for future research.

The scale is also limited by its factor structure. Each of the three factors (constructs) is unidimensional and comprises exactly four items each, which is the minimum for an overidentified model [28]. It would be interesting, in future research, to see if it is possible to supplement the models with more items and to find new models that measure other aspects of work with suicidal persons.

Another limitation regarding our interpretation is that although the three measurement models were found to be good fits over time, they might not be good fits across groups. Unfortunately, this has not been possible to analyze in the present study, due to the large variation in gender and in the number of respondents from each professional group.

## 7. Conclusions

This is the first study to examine items intended to measure various aspects of attitudes toward work with suicidal persons in (a) a heterogenous population and (b) with an analytic procedure that enables tests of the constructs of the models, as well as their validity and reliability. The resulting questionnaire was stable over time and may be used to measure attitudes toward suicide prevention, for instance, in studies evaluating suicide-preventive training programs, or to identify the impact of healthcare workers’ attitudes on the prevention of suicide.

## Figures and Tables

**Figure 1 ijerph-18-04001-f001:**
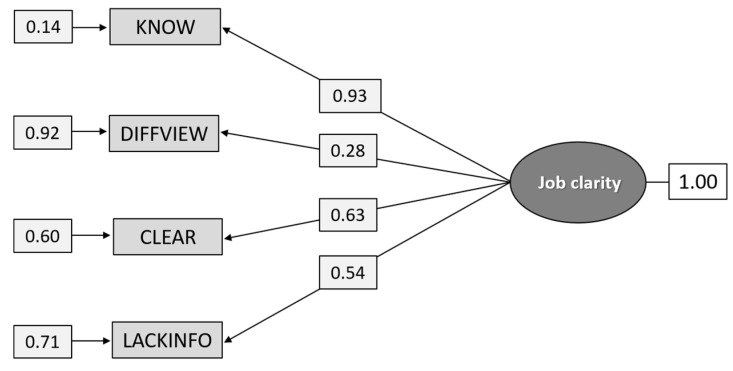
Path diagram: confirmatory factor analysis (CFA) of the construct Job clarity (n = 1.296).

**Figure 2 ijerph-18-04001-f002:**
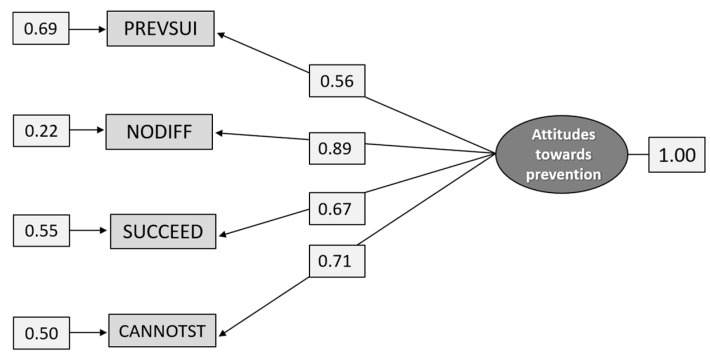
Path diagram: CFA of the construct Attitudes towards prevention, (n = 1.298).

**Figure 3 ijerph-18-04001-f003:**
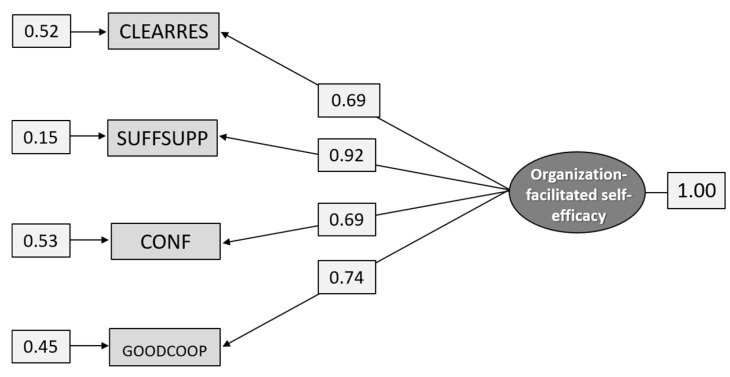
Path diagram: CFA analysis of the construct Organization-facilitated self-efficacy (n = 1.291).

**Figure 4 ijerph-18-04001-f004:**
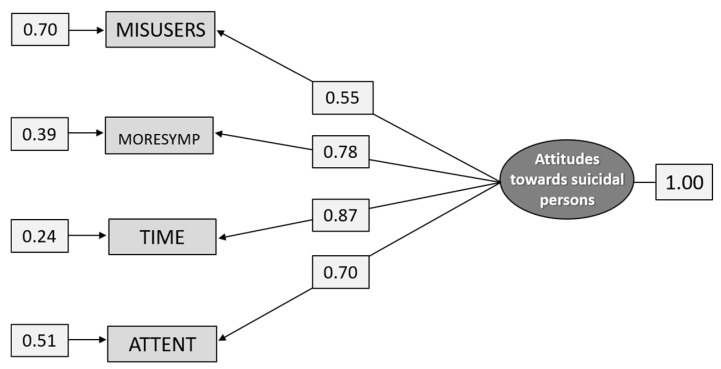
Path diagram: CFA analysis of the construct Attitudes toward suicidal persons (n = 1.271).

**Figure 5 ijerph-18-04001-f005:**
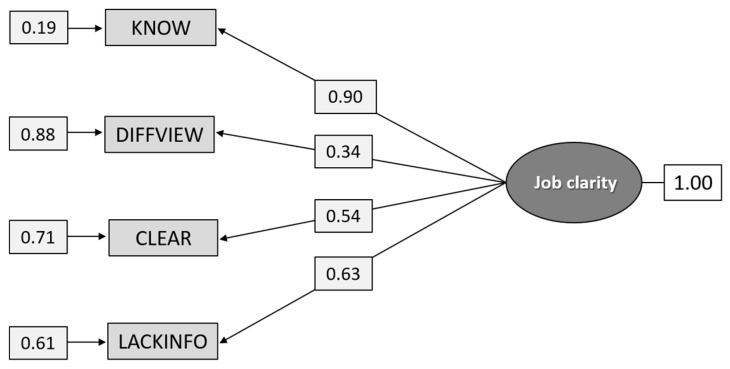
Path diagram: CFA analysis of the construct Job clarity (n = 630).

**Figure 6 ijerph-18-04001-f006:**
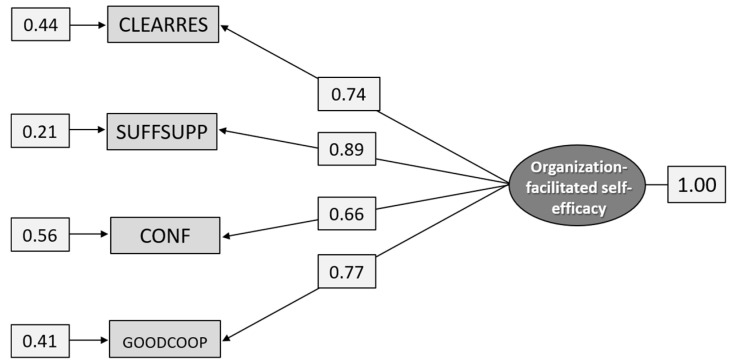
Path diagram: CFA analysis of the construct Organization-facilitated self-efficacy (n = 586).

**Figure 7 ijerph-18-04001-f007:**
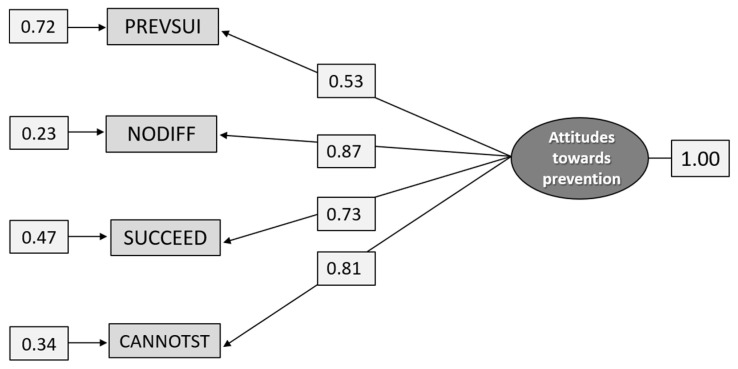
Path diagram: CFA analysis of the construct Attitudes toward prevention (n = 586).

**Figure 8 ijerph-18-04001-f008:**
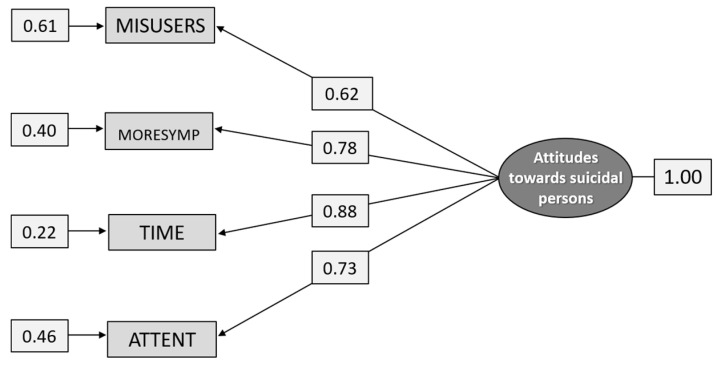
Path diagram: CFA analysis of the construct Attitudes toward suicidal persons (n = 615).

**Table 1 ijerph-18-04001-t001:** Constructs and items (cross-validation study).

Constructs	Items
A. *Job clarity* (The items that define the construct of *Job clarity* emanate originally from a study of Ahlberg-Hultén and Theorell [46].)	I know what is expected of me in work with suicidal individuals
	Different superiors have varying views on how and what I shall do in work with suicidal individuals
	I get clear and good instructions concerning management of care of suicidal individuals
	I lack knowledge and information about what is important in work with suicidal individuals
B. *Job confidence*	The division of responsibilities for risk assessment is clear and distinct
	I feel confident when working with suicidal individuals
	I have no one with whom I can share the responsibility for the suicidal individuals
	The co-operation concerning the suicidal individuals is well functioning
C. *Possibility to prevent suicides*	It is possible to prevent suicides
	It makes no difference what is done for suicidal individuals—they succeed sooner or later anyway
	If people really want to kill themselves, they will succeed despite receiving the best treatment
	Once people have made up their minds to commit suicide, you cannot stop them

**Table 2 ijerph-18-04001-t002:** Hypothesized measurement models.

Constructs	Items/Items
D. *Job support*	I have enough support in work with suicidal individuals
	I do not know what to do if somebody tells me that he/she want to kill himself/herself
	There are enough resources for me to do what is expected of me when I encounter suicidal persons
	Due to the ambiguous description of my responsibilities, I’m uncertain as to whether I should find out whether persons are suicidal or not
	I feel free to ask for help when I am not sure what to do in my work with suicidal individuals
E. *Attitudes toward suicidal persons*	Some suicidal individuals are abusing the health-care system
	I feel more sympathy for first time suicide attempters than for those who have made several attempts
	Suicidal persons often steal time from other persons with greater need for support and help
	Those who have made several suicide attempts do not really want to kill themselves
	Those who use non-lethal methods to attempt suicide only seek attention

**Table 3 ijerph-18-04001-t003:** Recommended cut-off criteria for fit indices.

Fit Index	Good Fit Values	Acceptable Fit Values
RMSEA	<0.05	<0.08
CFI	>0.97	>0.95
SRMR	<0.05	<0.08

**Table 4 ijerph-18-04001-t004:** Exploratory factor analysis (EFA) for ordinal data of items measuring work-related routines (n = 698).

Item	Factor 1	Unique Var
The division of responsibilities for risk assessment is clear and distinct [CLEARRES]	0.675	0.545
I feel confident when working with suicidal individuals [CONF]	0.714	0.490
I have no one with whom I can share the responsibility for the suicidal individuals [ALLRESP]	0.433	0.813
The co-operation concerning the suicidal individuals is well functioning [GOODCOOP]	0.769	0.409
I have enough support in work with suicidal individuals [SUFFSUPP]	0.902	0.186
I do not know what to do if somebody tells me that he/she want to kill himself/herself [NOTKNOW]	0.367	0.865
There are enough resources for me to do what is expected of me when I encounter suicidal persons [SUFFRES]	0.619	0.617
Due to the ambiguous description of my responsibilities, I’m uncertain as to find out whether persons are suicidal or not [UNCLEAR]	0.416	0.827
I feel free to ask for help when I am not sure what to do in my work with suicidal individuals [ASKHELP]	0.408	0.833

**Table 5 ijerph-18-04001-t005:** Validity, reliability, and model fit for subgroup G2 for the construct Organization-facilitated self-efficacy (OFSE), n = 605.

Items	Validity	Reliability
The division of responsibilities for risk assessment is clear and distinct [CLEARRES]	0.71	0.51
I have enough support in work with suicidal individuals [SUFFSUPP]	0.90	0.82
I feel confident when working with suicidal individuals [CONF]	0.65	0.42
The co-operation concerning the suicidal individuals is well functioning [GOODCOOP]	0.72	0.52
Model fit: χ^2^ = 0.216 [2], *p* = 0.8976, RMSEA = 0.0, *p* = 0.943, CI = 0.0–0.05, CFI = 1.00, SRMR = −0.00453		

**Table 6 ijerph-18-04001-t006:** Validity, reliability, and model fit for subgroup G2 for the construct Attitudes toward suicidal persons, n = 613.

Items	Validity	Reliability
Some suicidal individuals are abusing the health-care system [MISUSERS]	0.52	0.27
I feel more sympathy for first time suicide attempters than for those who have made several attempts [MORESYMP]	0.81	0.65
Suicidal persons often steal time from other persons with greater need for support and help [TIME]	0.85	0.72
Those who use non-lethal methods to attempt suicide only seek attention [ATTENT]	0.73	0.53
Model fit: χ^2^_[df = 2__]_ = 2.964, *p* = 0.2272, RMSEA = 0.0793, *p* = 0.126, [CI 0.0346–0.132], CFI = 0.999, SRMR = 0.0216		

**Table 7 ijerph-18-04001-t007:** Occupational Suicide Attitude Questionnaire (OSAQ-12).

**Attitudes toward prevention**	It is possible to prevent suicides It makes no difference what is done for suicidal individuals—they succeed sooner or later anyway If people really want to kill themselves, they will succeed despite receiving the best treatment Once people have made up their minds to commit suicide, you cannot stop them
**Attitudes toward suicidal persons**	Some suicidal individuals are abusing the health-care system I feel more sympathy for first time suicide attempters than for those who have made several attempts Suicidal persons often steal time from other persons with greater need for support and help Those who use non-lethal methods to attempts suicide only seek attention
**Organization-facilitated self-efficacy**	The division of responsibilities for risk assessment is clear and distinct I feel confident when working with suicidal individuals I have enough support in work with suicidal individuals The co-operation concerning the suicidal individuals is well functioning

## Data Availability

The data presented in this study are available on request from the corresponding author.

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
