# Peer review of "Measuring Attitudes toward Suicide Prevention among Occupational Staff Frequently Exposed to Suicidal Individuals: Psychometric Evaluation and Validation"

_ijerph, 2021, doi:10.3390/ijerph18084001_

Round 1

Reviewer 1 Report

Page 5, line 207: Exploratory not explorative, noted in other locations in the paper as well

Page 5, line 208: Confirmatory not confirmative

Page 6, line 222: authors state they use principal components analysis which is not EFA. These are two different types of data reduction techniques – one treating items as formative indicators and the other as reflective.

Page 6, line 222: what do the authors mean by “metric properties?”

Page 6, line 240: data are plural, so should read data are, not data is

In the figures: latent factors should be represented with ovals, not rectangles, this differentiates observed (rectangular) vs latent (oval) variables

Since the authors used a promax rotation, they are assuming the latent factors are correlated. However, when they run the CFA models, they run them for each factor individually. This contradicts their initial oblique (correlated) rotation. The model can have different fit if ran as separate latent factors versus all latent factors correlated in one model.

A little more detail of the validity and reliability analyses would be nice. Currently, there is one sentence discussing it. Also, it looks like the validity and reliability analyses were only conducted on some of the latent factors not all, clarification would be appreciated.

Page 13, line 438 (and later in discussion): I recommend to not use the word “invariant” or discuss invariant over time because the authors did not do any invariance testing.

Author Response

We would like to thank Reviewer 1 for these much valuable comments, and for constructive analytic insights. We hope that you will find our responses to be thorough, and our revisions satisfactory. We believe that they have contributed significantly to the quality of the paper.

Point 1: Page 5, line 207: Exploratory not explorative, noted in other locations in the paper as well

Response 1: We have changed the wording to “exploratory” in all instances.

Point 2: Page 5, line 208: Confirmatory not confirmative

Response 2: We have changed the wording to “confirmatory” in all instances.

Point 3: Page 6, line 222: authors state they use principal components analysis which is not EFA. These are two different types of data reduction techniques – one treating items as formative indicators and the other as reflective.

Response 3: We thank the reviewer for this very important comment which led to a lot of discussions in our research group, and ultimately to a rework of the data in the presented manuscript! Instead of using PCA, we now used the LISREL software to perform an EFA for ordinal data. These changes are now described in the methods section, and correspondingly in the results section. 

Point 4: Page 6, line 222: what do the authors mean by “metric properties?”

Response 4: In the revised paper we have defined/clarified the term “metric properties” as follows: “As the data consisted of ordinal data and without metric properties, i.e. that they are not continuous and thus not measured on a continuum or a scale, PRELIS [47] was used to estimate the polychoric correlations and their asymptotic covariance matrix when testing the above one-factor models [references 28, 44]. “

Point 5: Page 6, line 240: data are plural, so should read data are, not data is

Response 5: We have changed the wording to “data are” in all instances.

Point 6: In the figures: latent factors should be represented with ovals, not rectangles, this differentiates observed (rectangular) vs latent (oval) variables

Response 6: Thank you for pointing this out. It was a mistake on our part. The latent variables are now represented with ovals in all instances.

Point 7: Since the authors used a promax rotation, they are assuming the latent factors are correlated. However, when they run the CFA models, they run them for each factor individually. This contradicts their initial oblique (correlated) rotation. The model can have different fit if ran as separate latent factors versus all latent factors correlated in one model.

Response 7: Thank you for this comment. In the revised paper we now present the EFA for ordinal data. Since this analysis yielded a one-factor model, we believe this problem was resolved.

Point 8: A little more detail of the validity and reliability analyses would be nice. Currently, there is one sentence discussing it. Also, it looks like the validity and reliability analyses were only conducted on some of the latent factors not all, clarification would be appreciated.

Response 8: The reviewer makes a valid point. We have added more details concerning the validity and reliability analyses, both in the results presentation and in the discussion section.

Point 9: Page 13, line 438 (and later in discussion): I recommend to not use the word “invariant” or discuss invariant over time because the authors did not do any invariance testing.

Response 9: The word ”invariant” is now deleted and replaced by for instance “good fitting”. 

Reviewer 2 Report

Dear Authors,
I have read your work which I find very interesting. However, while I understand the scope of the study, I believe that some aspects need to be clarified
The sampling criterion (inclusion and exclusion criteria) would have been clarified. In particular if it was possible to detect the presence of psychiatric disorders in the participants and possibly how the data was managed.
Furthermore, it would also explain the different weight of the Gender. Females are much more. these elements should be put both in the introduction (e.g. suicides are more in women than in men, or because more women than men participate, etc.)
Furthermore, it should also be noted how to manage the data.
Then nothing is known about the prevention program in which the subjects participated. It is conceivable that the program affected the responses to the questionnaire. Already the difference in the numbers in answering between before and after the program makes a change evident.
In my opinion, these aspects should be clarified and placed in the analysis of the data as in the discussion.
For example, what is the relationship between the item and the prevention program?

Author Response

Thank you very much for your valuable comments on our paper. We hope that our clarifications will be satisfactory. As your three first comments are about the composition of the study population and the management of the data, we have chosen to answer them together.

Point 1: The sampling criterion (inclusion and exclusion criteria) would have been clarified. In particular if it was possible to detect the presence of psychiatric disorders in the participants and possibly how the data was managed.

Point 2: Furthermore, it would also explain the different weight of the Gender. Females are much more. these elements should be put both in the introduction (e.g. suicides are more in women than in men, or because more women than men participate, etc.)

Point 3: Furthermore, it should also be noted how to manage the data.

Response 1, 2, 3: The study population is composed of professionals that have participated in suicide-preventive training on a voluntary basis and all those who have answered to the electronic questionnaire and have worked with suicidal individuals at least once a month during the past six months were included in the study. Females are more predominant in these professional groups and there is thus a larger proportion of females than men in our representative sample. Due to the discrepancy between the number of females and men in the study we have not been able to analyze if the measurement models vary between gender. We have added a comment on this under the paragraph “Limitations” in the discussion section.

Point 4: Then nothing is known about the prevention program in which the subjects participated. It is conceivable that the program affected the responses to the questionnaire. Already the difference in the numbers in answering between before and after the program makes a change evident.

Response 4: We did not include information about the training program as this was not the scope of the present article. Note however, that while it’s possible that the training program affects attitudes, the focus of the present research is regarding the psychometric aspects of the tool used to measure attitudes. We analyze the items correlations to each other, whether they measure the same latent concept and if this is sustainable over time. Metaphorically, while medical treatment may improve depressive symptoms, it is unlikely that treatment will affect the statistical structure of the questionnaires measuring depression (like the Beck’s Depression Inventory). We thus believe that while training may improve attitudes, it does not change the psychometric properties of scales measuring attitudes. 

Point 5: In my opinion, these aspects should be clarified and placed in the analysis of the data as in the discussion.

For example, what is the relationship between the item and the prevention program?

Response 5: Please, see the above comments.